# Transcriptome Responses in *Medicago sativa* (Alfalfa) Associated with Regrowth Process in Different Grazing Intensities

**DOI:** 10.3390/plants13192738

**Published:** 2024-09-30

**Authors:** Dingyi Sun, Yalin Wang, Na Zhao

**Affiliations:** 1International Education College, Nanjing Forestry University, Nanjing 210037, China; 18801583522@163.com; 2Northwest Institute of Plateau Biology, Chinese Academy of Sciences, Xining 810008, China; 18437959165@163.com; 3University of Chinese Academy of Sciences, Beijing 100049, China

**Keywords:** grazing intensity, transcriptome analysis, plant–pathogen interaction pathway, plant hormone signal pathway

## Abstract

*Medicago sativa* L. (alfalfa), a perennial legume, is generally regarded as a valuable source of protein for livestock and is subjected to long and repeated grazing in natural pastures. Studying the molecular response mechanism of alfalfa under different grazing treatments is crucial for understanding its adaptive traits and is of great significance for cultivating grazing-tolerant grass. Here, we performed a transcriptomic analysis to investigate changes in the gene expression of *M. sativa* under three grazing intensities. In total, 4184 differentially expressed genes (DEGs) were identified among the tested grazing intensities. The analysis of gene ontology (GO) revealed that genes were primarily enriched in cells, cellular processes, metabolic processes, and binding. In addition, two pathways, the plant–pathogen interaction pathway and the plant hormone signal pathway, showed significant enrichment in the Kyoto Encyclopedia of Genes and Genomes (KEGG) analysis. Protein kinases and transcription factors associated with hormones and plant immunity were identified. The plant immunity-related genes were more activated under high grazing treatment, while more genes related to regeneration were expressed under light grazing treatment. These results suggest that *M. sativa* exhibits different strategies to increase resilience and stress resistance under various grazing intensities. Our findings provide important clues and further research directions for understanding the molecular mechanisms of plant responses to grazing.

## 1. Introduction

Grassland is a crucial element of the terrestrial ecosystem, playing a critical role in the support and regulation of ecological processes and providing habitats and foraging for grazing livestock [1]. Herbivore grazing is a primary practice that affects the structure and composition of grassland ecosystems [2]. Many experiments have been carried out to understand the effects of grazing on grassland ecosystems. Appropriate grazing can reduce the accumulation of dead material, accelerate ecosystem nutrient cycling, and increase the productivity of grasslands [3]. However, overgrazing could significantly affect plant growth and grassland regeneration, possibly even leading to grassland degradation [4]. Given the need for plants to quickly resume active photosynthesis and growth, the regrowth ability of plants is crucial after grazing [5].

For individuals, grazing can influence plants through morphological, physiological, or behavioral responses [6]. The most direct effects were plant morphology, including reduced plant height, decreased length of shoot internodes, and leaf area [7]. Long-term grazing may result in plant dwarfism, change the distribution pattern of above- and below-ground biomass, and lead to tolerance or avoidance strategies [7,8,9]. These characteristics eventually reduce the biomass of an individual plant and the whole community.

Past endeavors have found that the plant response to grazing lasts for 2 weeks or more, and that the response process changes over time [10]. The plant could mobilize all its resources to ensure its survival after being grazed. Through processes such as the reallocation of non-structural carbohydrates (NSCs) and interactions between various plant hormones, plants support leaf regrowth, restore photosynthetic tissues, and enhance self-defense [11,12]. Reactive oxygen scavenging systems have been observed to eliminate or minimize the toxicity of metabolic byproducts [9,13,14]. In addition, some plants produced defensive compounds, such as terpenes, flavonoids, and alkaloids, to resist grazing and pest damage [15]. Different grazing intensities also had diverse effects on plant characteristics, growth, and defense strategies [9]. Thus, we need a more comprehensive understanding of the adaptive changes that occur in plants after grazing.

Previous research has mainly focused on how plants adjust their morphological structure and physiological traits to cope with grazing disturbances, changing their traits to avoid or tolerate livestock foraging [16,17,18]. These adaptive changes are mainly influenced by factors such as the plant’s growth environment, species, and grazing intensity. Recent studies have highlighted that examining the molecular mechanism is essential for a comprehensive understanding of plant responses to grazing. For example, researchers have used transcriptome analysis to reveal major transcriptional changes in red clover during regrowth, finding several biotic and abiotic stress-related changes in plant growth and characterizing transcription-related protein families after mowing [19]. Another study compared two alfalfa populations with different grazing tolerance phenotypes and found differential expression in related pathways such as secondary metabolite production, primary carbohydrate metabolism, hormone signaling, and wound response [20]. Researchers have also used proteomic data to analyze the molecular mechanism of overgrazing-induced dwarfism in *Leymus chinensis* L. [21]. Although these studies have illustrated the molecular response mechanism of forage to grazing, transcriptional changes, adaptive strategies, and their interactions during forage regeneration under different grazing intensities are still largely unknown and need to be discovered.

*M. sativa*, one of the most important forage grasses, is known as the “king of forage grass” because of its high biological yield and nutritional value, good palatability, wide adaptability, and developed root system. It is cultivated as a fodder grass domestically and abroad [20,22]. *M. sativa* is a crucial species in natural grasslands and its proportion affects the nutrient status of the grassland. In contrast to cultivated varieties, wild *M. sativa* could be scavenged by grazing livestock and regenerated over several years. The regeneration process of *M. sativa* after grazing plays an important role in its survival and competition in natural grasslands. To further understand the long-term grazing response of *M. sativa*, we conducted transcriptome analysis on *M. sativa* after 2 weeks of grazing under three grazing treatments, identifying DEGs and enriched pathways between these treatments and non-grazing treatments. The present study is beneficial for enhancing the understanding of *M. sativa*’s response to grazing characteristics and providing scientific support for the rational utilization of this forage.

## 2. Materials and Methods

### 2.1. Plant Material and Grazing Treatments

In this study, we selected *M. sativa* from the perennial grazing experiment field in the Haibei Alpine Meadow Ecosystem Research Station managed by the Northwest Institute of Plateau Biology, Chinese Academy of Sciences (37°29′–37°45′ N, 101°12′–101°23′ E). The area of each grazing land plot was 9 m × 9 m, and the area of the fenced plot was 3 m × 15 m. The sample plots were established in 2009 and grazed by Tibetan sheep for 2 days per month during the vegetation growing season (June–September) every year [23]. Light, medium, and heavy grazing treatments (LG, MG and HG) were grazed by 3, 5, and 12 Tibetan sheep of uniform age and body condition, respectively, which were allowed to graze freely. Non-grazing (NG) plots are adjacent to these treatments and have the same terrain and soil conditions, and other influencing factors can be ignored. The vegetation consists mainly of *Carex alatauensis* S. R. Zhang, *Elymus nutans* Griseb., and *Poa pratensis* L., and the soil has been identified as alpine meadow soil [24].

New leaves of *M. sativa* were collected after grazing for two weeks at the beginning of September. The last grazing was in mid-August, which was the period of maximum plant biomass. To ensure the accuracy and consistency of sampling, 5 new and healthy leaves were randomly collected from three individual plants at 9:00 am to 10:00 am and mixed into one sample (15 leaves). The fresh leaves of *M. sativa* were immediately frozen in liquid nitrogen and later stored at −80 °C. Every treatment had three biological replicates.

### 2.2. RNA Extraction, Library Construction, and Data Processing

RNA was extracted from the leaves using Trizol regent (Invitrogen, Carlsbad, CA, USA). The quantity was evaluated using Nanodrop 2000 (NanoDrop, Wilmington, DE, USA) and the integrity was detected using the Agilent 2100 Bioanalyzer (Agilent Technologies, Inc, Santa Clara, CA, USA). RNA samples with a ratio of 260/280 nm greater than 1.8 and an RNA integrity number (RIN) greater than 7 were selected for further processes. The cDNA library was sequenced using the Illumina HiSeq 2000 platform.

Total RNA was extracted from the fresh leaves in 4 treated groups with 3 biological replicates. The original reads were cleaned by removing the reads containing adaptor sequences, low-quality reads (those with a quality score of Q < 20), and reads with an “N” percentage greater than 5%. Trinity software (version 2.6.6) was used to reassemble the transcriptome through three steps (Inchworm, Chrysalis, and Butterfly) [25].

### 2.3. Functional Annotation and Enrichment Analysis of DEGs

To annotate the unigene sequences of *M. sativa*, a BLASTx search was used to search against various databases (E-value < 10^−5^) [26,27], including the non-redundant protein (NR) database (http://www.ncbi.nlm.nih.gov (accessed on 4 November 2023)), the Swiss-Prot protein database (http://www.expasy.ch/sprot (accessed on 4 November 2023)), the Kyoto Encyclopedia of Genes and Genomes (KEGG) pathway database (http://www.genome.jp/kegg/ (accessed on 4 November 2023)), and the Cluster of Orthologous Groups (COG) database (http://www.ncbi.nlm.nih.gov/COG/ (accessed on 4 November 2023)).

RNA-Seq expression quantification was based on the fragments per kilobase per million mapped reads (FPKM) values. The false discovery rate (FDR) was calculated to correct multiple testing. The edgeR package (FDR < 0.05 and |log2FC| ≥ 1) was used to identify DEGs between LG-NG, MG-NG, and HG-NG. For the functional and pathway enrichment analyses, the DEGs were mapped to GO terms (*p*-value ≤ 0.05) and the KEGG database (*p*-value ≤ 0.05) [28].

### 2.4. qRT-PCR Validation

RNA was extracted from the four treatments of *M. sativa* from three independent biological replicates. Six mRNA sequences were randomly selected and genes with different expression patterns were verified using real-time qRT-PCR. Actin genes were used as reference genes to standardize the expression data. Appendix A lists the primer sequences. A LightCycler^®^ 480 II Real-time PCR Instrument (Roche, Switzerland) was used to perform Real-time PCR with a 10 μL PCR reaction mixture containing 1 μL of cDNA, 5 μL of 2× PerfectStartTM Green qPCR SuperMix, 0.2 μL of the forward primer, 0.2 μL of the reverse primer, and 3.6 μL of nuclease-free water. Reactions were incubated in a 384-well optical plate (Roche, Basel, Switzerland) at 94 °C for 30 s, followed by 45 cycles of 94 °C for 5 s and 60 °C for 30 s. Relative quantitative data were calculated using the 2^−ΔΔCT^ method [29]. All validations were performed in three biological and technical replicates.

## 3. Results

To understand the transcript profile of *M. sativa* after different grazing intensities, RNA-Seq libraries were constructed from new leaves, including three biological repeats for each treatment, and sequenced using Illumina paired-end sequencing technology. After removal of the adapter sequence, low-quality reads, and short reads, over 54 million clean reads were generated for leaf tissue. Q20 in all 12 samples was greater than 96% and the GC content was approximately 43%, indicating high-quality results (Table 1). Using Trinity software (Version 2.6.6), 23,3905 unigenes were assembled from all paired cleans, with a contig N50 of 799 bp and an average length of 553.41 bp (Table 2). The size distribution of the assembled unigenes showed that 95.19% (22,646) of the sequence length was between 0 to 2000 bp in length and the percentage of sequences >2000 bp was 4.91% (11,259). In total, 342,128 transcripts were generated. The N50 of transcripts was 1654 bp with an average length of 815.06 bp. There were 304,517 transcripts, of which 89.00% were less than 2000 bp in length (Appendix A).

### 3.1. Annotation and Classification of M. sativa

According to the Nr database (Appendix A), the E-value distribution indicated that 44.63% of the annotated unigenes had an E-value < 10^−30^ and 84.84% had high similarity (greater than 80%) with other species. The sequences showed the most similar BLASTx matches were to gene sequences of *Medicago truncatula* (15,188), followed by *Hordeum vulgare subsp. Vulgare* (2291) and *Trifolium subterraneum* (2246).

KOG classification was performed in the KOG database (Appendix A). A total of 40,036 unigenes were classified into 25 protein families. The highest category was ‘General function prediction only’ (3472), followed by ‘Posttranslational modification, protein turnover, chaperones’ (2924), ‘Translation, ribosomal structure, and biogenesis’ (2669), ‘Energy production and conversion’ (2069), and ‘Signal transduction mechanisms’ (1885). In addition, there were only five unigenes involved in the ‘Cell motility’ category.

According to the GO classification analysis (Figure 1), a total of 38,941 genes were identified and divided into three categories: biological process (BP), molecular function (MF), and cellular component (CC). In BP, the unique matching sequences were divided into 32 groups, with the highest number of ‘Cellular process’ in biological processes with 34,004 genes, followed by ‘metabolic process’ and ‘response to stimulus’ with 30,989 and 22,126 unigenes, respectively. CC was categorized into 20 groups, with ‘Cell’ and ‘Cell part’ being the two largest groups, containing 35,887 and 35,866 unigenes. For MF, unique sequences were divided into 15 classes. The largest subclass was ‘binding’ with 28,921 genes, followed by ‘catalytic activity’ with 24,282 genes.

Using the KEGG database, the identified unigenes were divided into different biochemical pathways (Figure 2). A total of 44,515 unigenes were grouped into five groups with a total of 35 sub-categories. Among the five groups, cellular process was the largest, with 22,203 unigenes assigned into 13 subgroups, followed by organismal system (7480), environmental information processing (6941), metabolism (4217), and genetic information processing (3674) into 10, 4, 5, and 3 subgroups, respectively. ‘Global and overview maps’ (7843) was the top in the 35 subgroups, followed by ‘Translation’ (3589), ‘signal transduction’ (3402), and ‘Carbohydrate metabolism’ (3221).

### 3.2. DEGs in Different Grazing Intensities

Compared with NG, 991, 1871, and 1322 DEGs were found in LG, MG, and HG, respectively (Figure 3). In our results, a total of 199 unigenes were co-expressed in three comparative manners, while 353 genes were co-expressed in LG-NG and MG-NG, 265 genes were co-expressed in LG-NG and HG-NG, and 582 genes were co-expressed in MG-NG and HG-NG. The types and numbers of co-expression genes may reflect the grazing intensity and responses of *M. sativa*.

### 3.3. Pathways Enrichment Analysis of DEGs

Compared with NG, we analyzed the top five enrichment pathways in the other three treatments (Table 3). In LG-NG, the five pathways were ‘plant–pathogen interaction’, ‘circadian rhythm–plant’, ‘glyoxylate and dicarboxylate metabolism’, ‘starch and sucrose metabolism’, and ‘fatty acid degradation’. MG-NG enriched five pathways involved in ‘plant–pathogen interaction’, ‘nitrogen metabolism’, ‘cutin, suberine, and wax biosynthesis’, ‘stilbenoid, diarylheptanoid, and gingerol biosynthesis’, and ‘plant hormone signal transduction’. Furthermore, the five pathways in HG-NG were ‘pentose and glucoronate interconversions’, ‘plant–pathogen interaction’, ‘oxidative phosphorylation’, ‘alpha-Linolenic acid metabolism’, and ‘plant hormone signal transduction’. The ‘plant hormone signal transduction’ pathway was enriched both in MG-NG and HG-NG, suggesting that plant hormones played significant roles in resistance to grazing stress. The ‘plant–pathogen interaction’ pathway was co-enriched in the LG, MG, and HG treatments, indicating that plants are vulnerable to pathogens after grazing and undergo a series of pathogenic resistance processes. Meanwhile, the other enriched pathways involved in the metabolism category indicated that the metabolism changes caused by grazing contributed to improving their grazing tolerance.

### 3.4. Plant Hormone Signal Transduction-Related DEGs after Grazing

According to KEGG enrichment analysis, the genes related to plant hormone signal transduction pathways were identified (Figure 4). The results showed a total of 45 DEGs, with 10 downregulated and 35 upregulated, including eight, 27, and 10 genes that were enriched in LG-NG, MG-NG, and HG-NG, respectively. Notably, we found three DEGs that were co-expressed in three comparative manners, including upregulated genes AUX1 and ARR-A and the downregulated gene AP2. In LG-NG, there were two ARR genes, three ethylene-related genes, and two auxin-related genes, which play a crucial role in ethylene and auxin biosynthesis and metabolism. In MG-NG, we found three SAUR and nine auxin-related genes in the auxin signal transduction pathway and nine genes involved in ethylene signaling. In HG-NG, we found two SAUR and two auxin-related genes in auxin signal transduction, three genes involved in ethylene signaling, and one gene, JAZ, involved in the jasmonic acid signal transduction pathway.

### 3.5. Plant–Pathogen Interaction Pathway-Related DEGs after Grazing

This pathway was significantly enriched in both up- and downregulated genes in three comparative manners with 49 genes (Figure 5). Our results showed that LG-NG had 13 genes, with five upregulated and eight downregulated genes, MG-NG had 18 genes, with eight upregulated and 10 downregulated genes, and HG-NG had 18 genes, with three upregulated and 15 downregulated genes. There were five co-expressed genes in three comparative manners, with four downregulated and one upregulated, which means *M. sativa* underwent gene expression changes after grazing and these changes were grazing intensity-dependent.

### 3.6. Analysis of Differentially Regulated Protein Kinases during Grazing Stress

The DEGs of protein kinases under different grazing intensities were identified. Compared to NG treatment, all 90 protein kinases were found in LG, MG, and HG treatments, with 51 DEGs upregulated and 39 DEGs downregulated, including 37 DEGs that were separately identified in LG-NG and MG-NG and 16 DEGs that were identified in HG-NG (Appendix A). The LRR receptor-like kinases (26.7%) and the Serine/Threonine kinase protein family (12.2%) accounted for a larger proportion. We also found several Ca^2+^-related kinases and wall-associated receptor kinases, with most genes upregulated in MG-NG and HG-NG, while no gene was found in LG-NG. Taken together, more receptor-like kinase genes were expressed in MG-NG and HG-NG.

### 3.7. Analysis of Differentially Regulated Transcription Factors (TFs) during Grazing Stress

TFs play an essential role in regulating gene expression during normal growth and development, as well as in response to different stresses. In our results, we found several TFs, including the AP2/ERF, NAC, WRKY, BHLH, MYB, and GATA transcription families (Appendix A). Eight, 43, and 15 related TFs were found in LG-NG, MG-NG, and HG-NG, respectively. The AP2/ERF family occupied the largest proportion of our results and was the most prevalent in MG-NG. The NAC family was the second most common of the upregulated DEGs in MG-NG, and MYB TFs were found in three grazing treatments.

### 3.8. Experimental Validation

Six DEGs were selected for qRT-PCR analysis under different grazing intensities (Figure 6). These candidate genes involved in two auxin signal transduction-related proteins (TRINITY_DN377_c0_g1 and TRINITY_DN6020_c0_g1), a receptor-like kinase (TRINITY_DN2832_c1_g2), an AP2/ERF and B3 domain transcription factor (TRINITY_DN2394_c0_g1), a heat shock transcription factor (TRINITY_DN5072_c0_g1), and a cyclic nucleotide-gated ion channel protein (TRINITY_DN5232_c0_g2). These analyses supported similar trends in expression patterns observed by RNA-seq for six genes under the different grazing conditions, despite differences in gene expression in individual treatments.

## 4. Discussion

Grazing is an essential use of grassland and a common pressure for pasture and grassland plants. *M. sativa* is an important species in alpine meadows, known for its high nutrition and its ideal forage quality [30]. Under grazing conditions, plants could change their growth strategies to adjust to grazing damage, such as changing its photosynthetic activity, altering the growth rate, or enhancing defense capacity [9]. However, there are fewer studies on the effects of different grazing intensities on transcriptional activity in grassland plants. In our study, we sequenced 12 libraries prepared from NG, LG, MG, and HG group samples using Illumina RNA-Seq technology and obtained high-quality de novo assembly data. A total of 233,905 unigenes were annotated through Trinity software (Version 2.6.6), and the Nr databases showed that the most similar species was *M. truncatula* because *M. sativa* had high genetic similarity to *M. truncatula*. Meanwhile, the KEGG annotation analysis showed that most of the genes were enriched in ‘Translation’, ‘Signal transduction’, ‘Carbohydrate metabolism’, and ‘Energy metabolism’, which means that after grazing, *M. sativa* underwent an active change to adjust to grazing damage. According to the KEGG enrichment analysis, under different grazing intensities, the plant hormone signal transduction pathway and the plant–pathogen interaction pathway are co-expressed, which may enhance the adaptability of *M. sativa* to grazing.

Plant hormones are not only important for plant growth and development activities but also play an indispensable role as signaling factors in defense and immune responses [15,31]. Plant hormones often exert interdependent effects through complex antagonistic or synergistic interactions to control plant immunity or growth [32]. In our results, the auxin- and ethylene-related signal transduction genes had significantly different expressions, which means that the auxin and ethylene hormones were activated to promote regrowth and defend against the harm from grazing. Auxin is often associated with cell expansion and growth-promoting processes, and ethylene is thought to be an important component of the plant’s immune response to pathogens [33]. In the LG-NG and MG-NG groups, the auxin-related genes occupied a large proportion, while the ethylene-related genes had a greater proportion in HG-NG, which means the auxin-related genes were more active than ethylene-related genes in the LG and MG treatments, while HG treatment shows an opposite trend.

Wounding is one of the major threats to plant survival after grazing because it not only damages plants’ tissues but also provides a site for potential pathogen entry [34]. After grazing, the first defense barriers of the plant, such as the cuticle, cell wall, and produced anti-microbial compounds, are breached, and the immune system is activated with a complex array of responses. In our results, we found multiple genes involved in plant defense against pathogens. Ca^2+^-related genes are signal sensors that transduce calcium signals to downstream biological events [35], and the cell wall-associated receptor kinases identified in MG and HG treatments are reported to be required for cell expansion and participate in the response to pathogens and stress [36]. The LRR receptor-like proteins, Serine/Threonine kinase family proteins, and receptor-like kinases play important roles in the plant’s immune system and several genes were differentially expressed compared to NG treatment [37,38]. Meanwhile, these genes were more active in MG and HG, while fewer related genes were identified in LG in our results.

Previous studies have suggested that compensatory growth occurs after grazing and plants showed different compensatory characteristics depending on grazing intensity [39]. Studies on the Inner Mongolian grassland showed that the above-ground productivity of vegetation recovered rapidly after super-light grazing and showed equal compensatory growth, and light and moderate grazing promoted the above-ground biomass of vegetation, which showed over-compensatory growth, while after heavy grazing, the above-ground productivity was difficult to recover and showed under-compensatory growth [40]. These findings may be explained by our results. In the LG treatment, the genes involved in the regeneration process were more active than those involved in the defense process, while in the HG treatment, genes associated with the defense process were more active. Recent studies have shown that plants have mixed tolerance and resistance defense mechanisms, but there is a trade-off between the two strategies, whereby selection for increased resistance may lead to decreased tolerance, resulting in a diversity of plant phenotypes after grazing [15]. Meanwhile, the defense strategies of plants against herbivores can respond to different levels of consumption, with some characteristics being induced only under high herbivore conditions and others being induced under moderate herbivore conditions [41]. In our results, the DEGs expressed in three grazing intensities were similar to these studies, while the limitation of our results is that the physiological and morphological characteristics of the plant were not studied. Thus, the response of *M. sativa* to different grazing intensities should be further analyzed in terms of physiology, morphology, and transcription.

## 5. Conclusions

This study used transcriptomic methods to compare and analyze mRNA expression in *M. sativa* leaves under different grazing intensities. A total of 4184 DEGs were identified through pairwise comparisons among different grazing treatments. Further analysis revealed that the DEGs were mainly found in the plant–pathogen interaction and hormone signal transduction pathways, along with related kinase proteins and transcription factors. Meanwhile, the genes expressed in the HG treatment were mostly involved in the defense process, while the genes expressed in the LG treatment were mostly involved in the regeneration process. These results have expanded our understanding of the molecular mechanisms in *M. sativa* under different grazing intensities, which will provide directions and theoretical support for further research on the *M. sativa* response to grazing stress.

## Figures and Tables

**Figure 1 plants-13-02738-f001:**
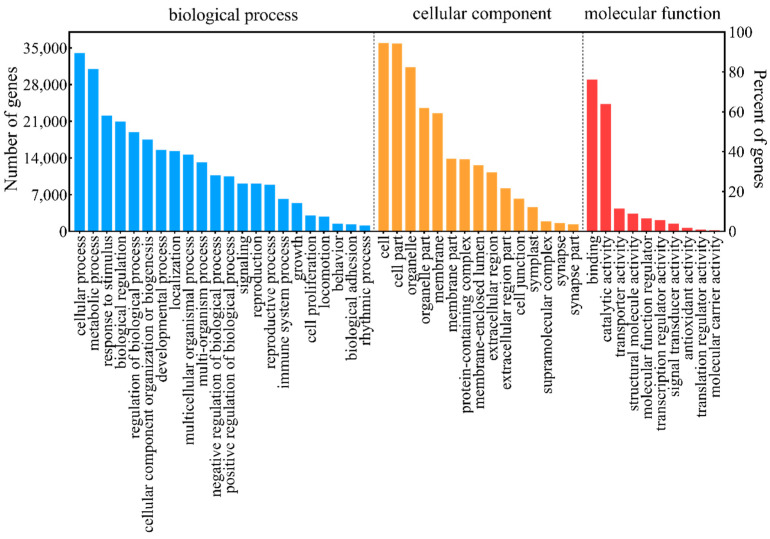
GO function classification of unigenes for *M. sativa*. The y-axis indicates the number (left) and percent (right) of annotated unigenes in each term. Blue histograms represent “biological process”, yellow histograms represent “cell component”, and red histograms represent “molecular function”.

**Figure 2 plants-13-02738-f002:**
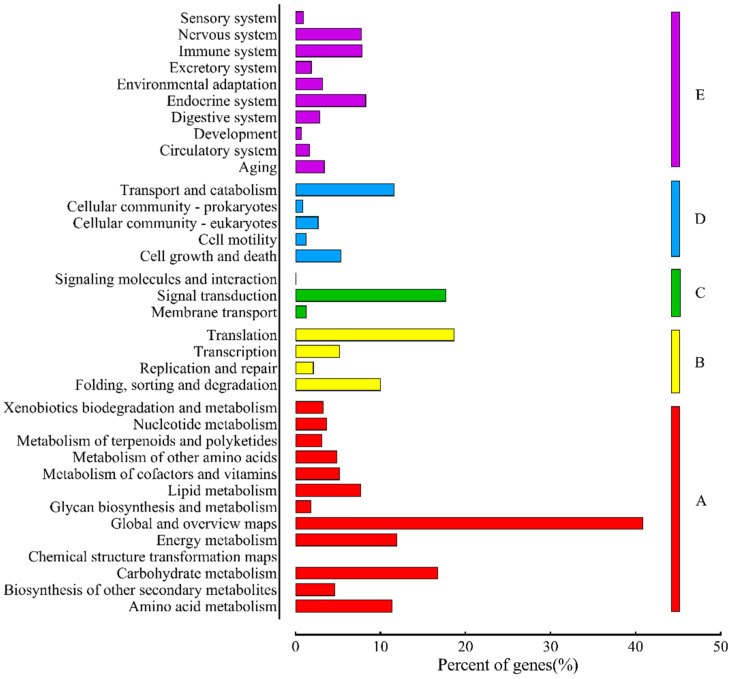
KEGG function classification for *M. sativa.* The y-axis indicates different KEGG categories and the x-axis indicates the percent of KEGG genes. A: Cellular Process, B: Environmental Information Processing, C: Genetic Information Processing, D: Metabolism, and E: Organismal Systems.

**Figure 3 plants-13-02738-f003:**
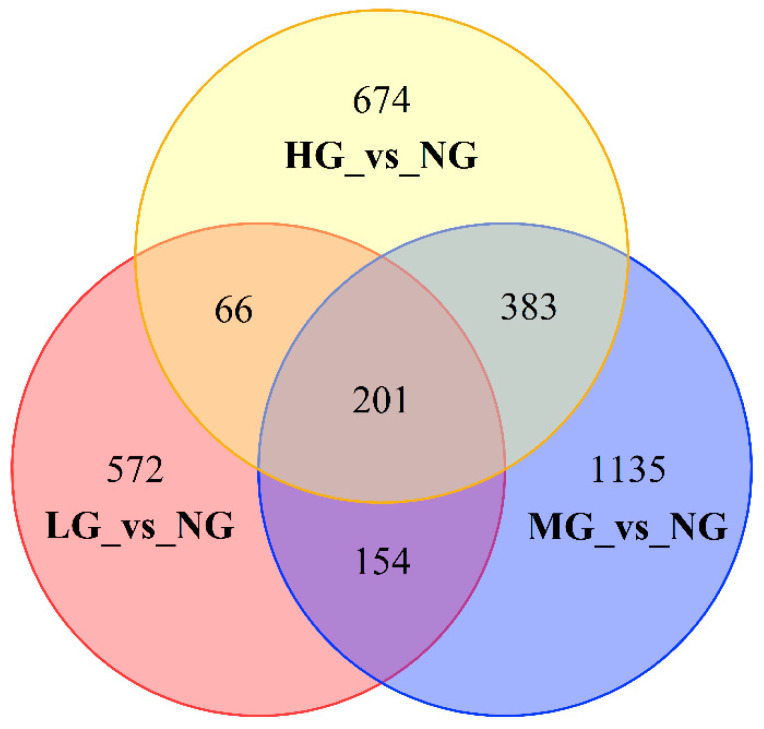
Venn diagram of the functional annotation, where each number represents the number of differentially expressed genes (DEGs) in different grazing intensities. Red indicates the DEGs under light grazing and non-grazing treatments; blue indicates the DEGs under moderate grazing and non-grazing treatments; and yellow indicates the DEGs under heavy grazing and non-grazing treatments.

**Figure 4 plants-13-02738-f004:**
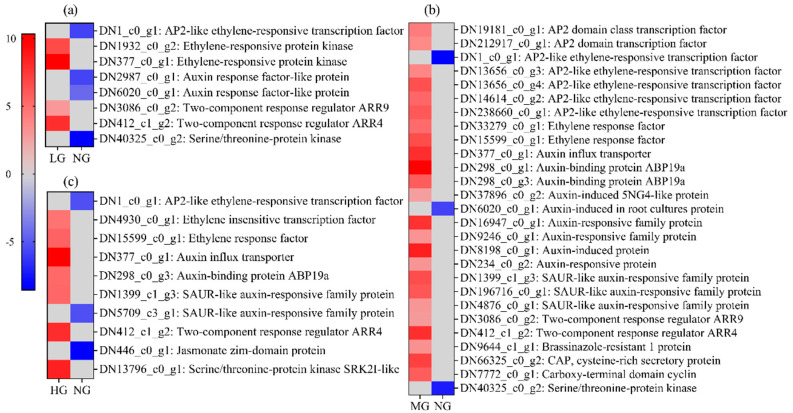
Plant hormone signal transduction-related DEGs in different grazing intensities. (**a**), (**b**), and (**c**) represent LG-NG, MG-NG, and HG-NG treatment, respectively. Red, blue, and gray denote upregulation, downregulation, and no change in expression, respectively.

**Figure 5 plants-13-02738-f005:**
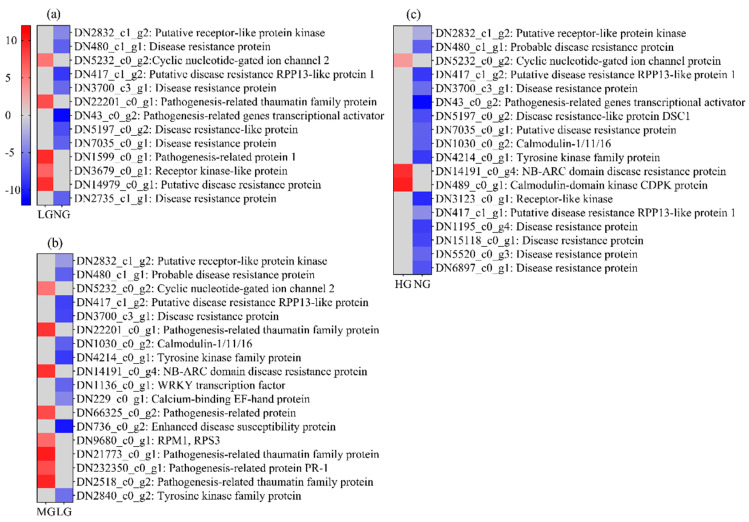
Plant–pathogen interaction pathway-related DEGs in different grazing intensities. (**a**), (**b**), and (**c**) represent LG-NG, MG-NG, and HG-NG treatment, respectively. Red, blue, and gray denote upregulation, downregulation, and no change in expression, respectively.

**Figure 6 plants-13-02738-f006:**
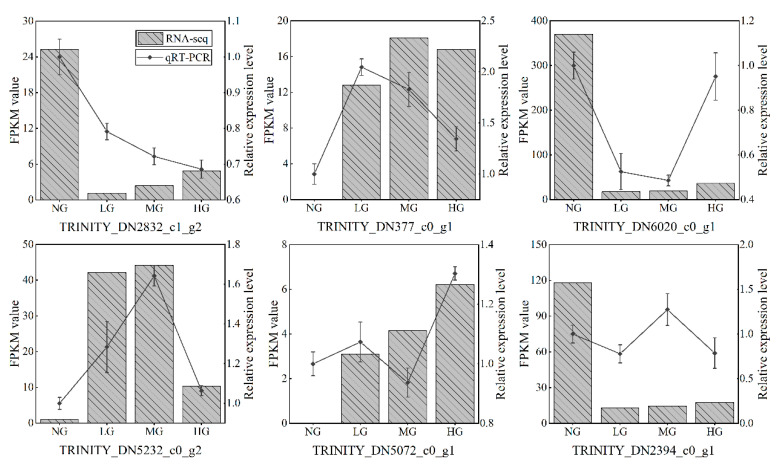
Comparison between the results of qRT-PCR and RNA-seq analyses of selected DEGs. The x-axis represents four treatments involved in non-grazing (NG), light grazing (LG), moderate grazing (MG), and heavy grazing (HG). The left y-axis and the right y-axis indicate the fragments per kilobase per million mapped reads (FPKM) value and the gene’s relative expression levels, respectively.

**Table 1 plants-13-02738-t001:** Sequencing the *M. sativa* transcriptome in twelve samples from plants that underwent light grazing treatment (LG-1, LG-2, and LG-3), moderate grazing treatment (MG-1, MG-2, and MG-3), heavy grazing treatment (HG-1, HG-2, and HG-3), and non-grazing treatment (NG-1, NG-2, and NG-3).

Sample	Clean Reads	Error%	Q30%	Q20%	GC%
LG-1	37,879,600	0.0272	92.17	97.11	42.41
LG-2	42,037,020	0.0274	92.03	97.03	42.67
LG-3	37,761,718	0.0279	91.62	96.80	43.14
MG-1	52,233,430	0.0272	92.18	97.12	43.15
MG-2	37,810,516	0.0275	91.99	96.96	42.78
MG-3	41,746,140	0.0269	92.40	97.23	42.74
HG-1	46,232,500	0.0272	92.26	97.09	42.85
HG-2	51,027,360	0.0271	92.13	97.16	43.05
HG-3	46,626,312	0.0281	91.43	96.76	43.29
NG-1	48,192,934	0.0267	92.63	97.31	43.20
NG-2	50,844,286	0.0274	91.98	97.04	42.67
NG-3	48,072,326	0.0271	92.26	97.16	42.95

**Table 2 plants-13-02738-t002:** Statistics of sequencing and assembly results.

Type	Unigene	Transcripts
Total sequence number	233,905	342,128
Percent GC	43.59	41.08
Largest	21,380	21,380
Smallest	201	173
Average	553.41	815.06
N50	799	1654
N90	239	276

**Table 3 plants-13-02738-t003:** Significantly enriched gene pathways following the different grazing intensities.

Pathway	*p*-Value	Pathway ID	Category
LG-NG			
Plant–pathogen interaction	0.018	ko04626	Organismal system
Circadian rhythm–plant	0.020	ko04712	Organismal system
Glyoxylate and dicarboxylate metabolism	0.027	ko00630	Metabolism
Starch and sucrose metabolism	0.034	ko00500	Metabolism
Fatty acid degradation	0.044	ko00071	Metabolism
MG-NG			
Plant–pathogen interaction	0.000	ko04626	Organismal system
Nitrogen metabolism	0.001	ko00910	Metabolism
Cutin, suberine, and wax biosynthesis	0.004	ko00073	Metabolism
Stilbenoid, diarylheptanoid, and gingerol biosynthesis	0.012	ko00945	Metabolism
Plant hormone signal transduction	0.026	ko04075	Environmental information processing
HG-NG			
Pentose and glucoronate interconversions	0.002	ko00040	Metabolism
Plant–pathogen interaction	0.005	ko04626	Organismal system
Oxidative phosphorylation	0.008	ko00190	Metabolism
Alpha-Linolenic acid metabolism	0.022	ko00592	Metabolism
Plant hormone signal transduction	0.025	ko04075	Environmental information processing

## Data Availability

The original contributions presented in the study are included in the article and Appendix A. The raw sequence data reported in this paper have been deposited in the National Center for Biotechnology Information (PRJNA1151746), publicly accessible at https://www.ncbi.nlm.nih.gov/ (accessed on 5 September 2024).

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
