# Peer review of "Transcriptome Responses in Medicago sativa (Alfalfa) Associated with Regrowth Process in Different Grazing Intensities"

_plants, 2024, doi:10.3390/plants13192738_

Round 1

Reviewer 1 Report

Comments and Suggestions for Authors

I review the manuscript titled “Transcriptome responses in Medicago sativa associated with re-growth process in different grazing intensities” by Sun et al. Authors explore transcriptome analysis to unraveled genes mediating the molecular mechanisms of plant response to grazing.

Overall, the research concept is scientific. Nevertheless, some corrections are required.

Specific comments:

1) The plagiarism report of the MS is too high at 38%. Authors should carefully work on the MS to reduce the plagiarism to a bearable number ≤ 10%.

2) There is a need to check the English language and rephrase some sentences in the MS.

2) The statement problem and justification of the research should be well stated in the Introduction section and end the Introduction with a concluding statement.

3) The discussion can be improved with the recent information/citations relating to the study.

5) L365: The links to the data information should be provided in the ‘Data Availability Statement” section.

Minor comments.

L2 and 9: add the common name to the scientific name

L10-13: Rephrase this sentence

L14: 4,184

L18: write in full and bracket KEGG at the first time of mentioning

L18-19: rephrase

L20, 21: LG, HG, write the full meaning

L32-33: rephrase

L34: materials…

L61, 64, 67: the references should be written appropriately following the journal author guide.

L86: provide the coordinates

L91-92: add the common names

L93: rephrase                                    

Be consistent is writing this as “Medicago sativa L.” throughout the MS

In Tables 1 and 2, L158-184: the numbers stated should be separated with comma where appropriate.

Comments on the Quality of English Language

Need some minor checks

Author Response

Comments and Suggestions for Authors

I review the manuscript titled “Transcriptome responses in Medicago sativa associated with re-growth process in different grazing intensities” by Sun et al. Authors explore transcriptome analysis to unraveled genes mediating the molecular mechanisms of plant response to grazing.

Overall, the research concept is scientific. Nevertheless, some corrections are required.

Specific comments:

1) The plagiarism report of the MS is too high at 38%. Authors should carefully work on the MS to reduce the plagiarism to a bearable number ≤ 10%.

2) There is a need to check the English language and rephrase some sentences in the MS.

3) The statement problem and justification of the research should be well stated in the Introduction section and end the Introduction with a concluding statement.

4) The discussion can be improved with the recent information/citations relating to the study.

5) L365: The links to the data information should be provided in the ‘Data Availability Statement” section.

Thank you very much for taking the time to review this manuscript. I have modified the repetition rate of the article as requested, but the repetition rate in the method is still relatively high, which seems inevitable. And the sentences requested to be rephrase has been modified and marked red in the article. I have modified the “Data Availability Statement” section, but I am not sure how to obtain the link for supplementary material, and whether it will be available after the article is published.

Minor comments.

L2 and 9: add the common name to the scientific name

It has been modified as required and marked red in the article.

L10-13: Rephrase this sentence

It has been modified as required and marked red in the article.

L14: 4,184

It has been modified as required and marked red in the article.

L18: write in full and bracket KEGG at the first time of mentioning

It has been modified as required and marked red in the article.

L18-19: rephrase

It has been modified as required and marked red in the article.

L20, 21: LG, HG, write the full meaning

It has been modified as required and marked red in the article.

L32-33: rephrase

It has been modified as required and marked red in the article.

L34: materials…

It has been modified as required and marked red in the article.

L61, 64, 67: the references should be written appropriately following the journal author guide.

It has been modified as required and marked red in the article.

L86: provide the coordinates

It has been modified as required and marked red in the article.

L91-92: add the common names

There is no common name for plants, the full Latin name of plants is added in the text.

L93: rephrase                                    

It has been modified as required and marked red in the article.

Be consistent is writing this as “Medicago sativa L.” throughout the MS

In Tables 1 and 2, L158-184: the numbers stated should be separated with comma where appropriate.

It has been modified as required and marked red in the article.

Reviewer 2 Report

Comments and Suggestions for Authors

The authors looked at transcriptomic changes in response to different grazing intensities. An important field of study.

Main concerns:

1.       The authors need to better describe the sampling and grazing treatments. Since the experimental fields used have a mixed vegetation, and sheep generally prefer legumes, it is important that the authors report the coverage for each crop species. The authors also need to report the grazing procedure: When were the fields grazed? For how long were the fields grazed? Ie, was the grazing intensity achieved by reducing the amount of time the sheep were allowed to graze, or by modifying the number of sheep, as this may affect how each field was grazed, plot size, number of grazers (or grazer densitiy). How was grazing intensity quantified? With respect to sampling, the authors need to better describe how was sampled: Especially under light grazing, many plants will not have been touched. What was the sample size (ie leaves from how many individual plants)? How was randomization established? What does (line 96) ‘in the same light direction’ mean?

2.       The authors did a rudimentary transcriptomics analysis that limited the amount of useful information the authors extracted from the analysis to: grazing induces a resistance response and hormonal changes. This is known already and therefore the impact of this study is limited.

Specific comments:

English language reasonably ok, but still needs some work.

Line 128- ‘noemalize’

Line 322- …and the ‘cell’ wall-associated receptor kinase…

Figure 1 could be supplementary and is difficult to read

The point of figure 2 is not clear

Figures 3 and 4: increase font size and could go into supplementary figures

Figure 5: state if same samples were used and add statistical analysis to the figures

Table 1 and two can go to supplementary data

Comments on the Quality of English Language

Use of English language is reasonable, but some good editing is still required to make it up to standard.

Author Response

The authors looked at transcriptomic changes in response to different grazing intensities. An important field of study.

Main concerns:

  1. The authors need to better describe the sampling and grazing treatments. Since the experimental fields used have a mixed vegetation, and sheep generally prefer legumes, it is important that the authors report the coverage for each crop species. The authors also need to report the grazing procedure: When were the fields grazed? For how long were the fields grazed? Ie, was the grazing intensity achieved by reducing the amount of time the sheep were allowed to graze, or by modifying the number of sheep, as this may affect how each field was grazed, plot size, number of grazers (or grazer densitiy). How was grazing intensity quantified? With respect to sampling, the authors need to better describe how was sampled: Especially under light grazing, many plants will not have been touched. What was the sample size (ie leaves from how many individual plants)? How was randomization established? What does (line 96) ‘in the same light direction’ mean?

Thank you for your comments. I have added the content in the materials and methods of the article and marked it in red.

  1. The authors did a rudimentary transcriptomics analysis that limited the amount of useful information the authors extracted from the analysis to: grazing induces a resistance response and hormonal changes. This is known already and therefore the impact of this study is limited.

Specific comments:

English language reasonably ok, but still needs some work.

Line 128- ‘noemalize’

It has been modified as required and marked red in the article.

Line 322- …and the ‘cell’ wall-associated receptor kinase…

It has been modified as required and marked red in the article.

Figure 1 could be supplementary and is difficult to read

The point of figure 2 is not clear

Figures 3 and 4: increase font size and could go into supplementary figures

Figure 5: state if same samples were used and add statistical analysis to the figures

Table 1 and two can go to supplementary data

The pictures and tables have been modified in the article, and the original pictures are attached

Reviewer 3 Report

Comments and Suggestions for Authors

The authors investigated changes in the gene expression of M. sativa under different grazing intensities by transcriptomic analysis. The Manuscript is well written and easy to follow. It represents an in-depth analysis of plant responses to grazing and confirms previous studies. I have just a couple of questions for the authors. The first is: how can you say the level of grazing (30%, 45%, 60%)? Perhaps how long the sheep graze on the field? Please, explain it on Material and Methods. The second is: considering the results you have obtained, can your study provide the right direction to the farmers, or does it simply provide information for future studies? You could enrich the Conclusion considering this aspect.  

Author Response

The authors investigated changes in the gene expression of M. sativa under different grazing intensities by transcriptomic analysis. The Manuscript is well written and easy to follow. It represents an in-depth analysis of plant responses to grazing and confirms previous studies. I have just a couple of questions for the authors. The first is: how can you say the level of grazing (30%, 45%, 60%)? Perhaps how long the sheep graze on the field? Please, explain it on Material and Methods. The second is: considering the results you have obtained, can your study provide the right direction to the farmers, or does it simply provide information for future studies? You could enrich the Conclusion considering this aspect.

Thank you for your comments. I have added the content in the materials and methods and conclusions of the article and marked it in red.

Round 2

Reviewer 2 Report

Comments and Suggestions for Authors

The authors did some changes accordingly to my concern about the details of sampling and grazing treatment. However, some of the concerns were not addressed such as 

 'With respect to sampling, the authors need to better describe how was sampled: Especially under light grazing, many plants will not have been touched. What was the sample size (ie leaves from how many individual plants)? How was randomization established? What does (line 96) ‘in the same light direction’ mean?'

 Even with the revised information, there are still some issues with the sample collection as described in the manuscript. For example, the samples were randomly collected from 3 different plants without mentioning how many leaves were collected. Samples from 3 plants is insufficient to represent the whole population in the treatment.

 For the specific comments, the authors did revise the figures as mentioned although most of the figures were kept in the manuscript and not moved to the supplementary. Figure 5 that was changed to figure 8 still has no statistical analysis and the qPCR validation test only poorly reflects the RNA-seq data.

 Also the concern about the limited amount of useful information the authors extracted from the analysis: The authors did not address this concern in the revised manuscript which I think is quite important.

Comments on the Quality of English Language

Use of the English language is ok, but some thorough editing by a native English speaker is still required.

Author Response

Dear reviewer:

Thank you for your valuable comments. The following is my answer, please check.

First of all, regarding the details of the sampling and grazing treatment. Our experimental plot was constructed in 2009. Plants in the sample plots with different grazing intensities were affected by the same grazing intensity over a longer period of time. The growth status of plants in different grazing intensity plots is relatively consistent. Therefore, all plants in these plot should have been affected by grazing and there should be no phenomenon of being untouched under light grazing. Meanwhile, the samples we collected should also be sufficient to represent the growth status in the sample grazing plot. We collected 5 leaves per plant. One sample was mixed for every 3 individual plant leaves (15 leaves). 3 replicates per grazing treatment and these samples were collected form 3 randomly selected orientations out of the 4 directions in the sample plot. 'In the same light direction' was taken into account at the time of sampling that light orientation might have an effect on the results, so the sample leaves were collected with the same light direction as far as possible during the sample leaf collection process.

Second, the figure in the text have been reduced and added to supplement files. The figure 8 has also been analyzed statistically.

Third, other authors and I have seriously considered your suggestions on extracting useful information from the paper, and hope to further improve them in future experiments.

Thank you again for your valuable suggestions on the paper.
